# Uncoupling of dynamin polymerization and GTPase activity revealed by the conformation-specific nanobody dynab

Valentina Galli[1], Rafael Sebastian[2], Sandrine Moutel[3,4], Jason Ecard[3], Franck Perez[3], Aurélien Roux[1,5]*

[1]Department of Biochemistry, University of Geneva, Geneva, Switzerland; [2]Department of Computer Sciences, Universidad de Valencia, Valencia, Spain; [3]Institut Curie, PSL Research University, Paris, France; [4]Translational Department, Institut Curie, Paris, France; [5]Swiss National Centre for Competence in Research Programme Chemical Biology, Geneva, Switzerland

**Abstract** Dynamin is a large GTPase that forms a helical collar at the neck of endocytic pits, and catalyzes membrane fission (Schmid and Frolov, 2011; Ferguson and De Camilli, 2012). Dynamin fission reaction is strictly dependent on GTP hydrolysis, but how fission is mediated is still debated (Antonny et al., 2016): GTP energy could be spent in membrane constriction required for fission, or in disassembly of the dynamin polymer to trigger fission. To follow dynamin GTP hydrolysis at endocytic pits, we generated a conformation-specific nanobody called dynab, that binds preferentially to the GTP hydrolytic state of dynamin-1. Dynab allowed us to follow the GTPase activity of dynamin-1 in real-time. We show that in fibroblasts, dynamin GTP hydrolysis occurs as stochastic bursts, which are randomly distributed relatively to the peak of dynamin assembly. Thus, dynamin disassembly is not coupled to GTPase activity, supporting that the GTP energy is primarily spent in constriction.
DOI: https://doi.org/10.7554/eLife.25197.001

*For correspondence:
aurelien.roux@unige.ch

Competing interests: The authors declare that no competing interests exist.

## Introduction

There are three mammalian variants of dynamin that share 80% sequence identity. They differ in their expression (*Schmid and Frolov, 2011*; *Ferguson and De Camilli, 2012*): dynamin-1 is specific to neurons, dynamin-2 is ubiquitously expressed, dynamin-3 is located in the brain, lungs and testis (*Vaid et al., 2007*). Dynamins are constituted of 5 domains: the GTPase (G) domain, the stalk – a rigid coiled-coil, which mediates assembly into helical polymers through 3 structurally conserved interfaces (*Faelber et al., 2011*; *Ford et al., 2011*; *Reubold et al., 2015*), the Bundle Signaling Element (BSE), which connects the GTPase domain to the stalk (*Chappie et al., 2010*; *Chappie et al., 2011*), a Pleckstrin Homology (PH) domain, which binds to phosphatidylinositol 4,5-bisphosphate (PI(4,5)P$_2$) (*Salim et al., 1996*) and the proline-rich domain (PRD), which binds partners such as Bin Amphiphysin Rvs (BAR)-domain proteins (*Raimondi et al., 2011*).

Dynamin can assemble into a helical coil around membrane necks and tubes. The constriction of the coil is required for fission and the most constricted state of dynamin is observed in presence of GTP (*Antonny et al., 2016*). However, the use of GTP energy is still debated: either it is used in helix constriction through twisting, or in disassembly of the constricted polymer (*Antonny et al., 2016*). When GTP is added to pre-assembled dynamin tubes in vitro, rapid torsion and constriction are observed (*Roux et al., 2006*). When GTP is added along with dynamin to membranes in vitro, highly constricted helices are observed, suggesting that GTP-bound dynamin assembles directly into a highly constricted state (*Sundborger et al., 2014*). However, the K44A used in this study has

residual GTPase activity, and thus the highly constricted state could result from limited GTP hydrolysis. Furthermore, a partial depolymerization is observed when GTP is added to dynamin tubules, which could play a role in fission (*Bashkirov et al., 2008*; *Pucadyil and Schmid, 2008*). This depolymerization may come from the fact that the PH domain of dynamin in the most-constricted state is titled, probably breaking the bond between dynamin and the membrane (*Sundborger et al., 2014*). It is thus difficult to infer from in vitro data how GTP hydrolysis, constriction and disassembly are coupled.

Structural studies have neither solved whether fission occurs by active constriction or active depolymerization of the dynamin helix. In the presence of GTP or GTP-analogs, GTPase domains form dimers (called G-G), explaining the higher propensity of dynamin to self-assemble with GTP. However, the strongest GTPase domain dimerization occurs using GDP·AlF$_4$, that mimics the transition state of hydrolysis (*Chappie et al., 2010*; *Chappie et al., 2009*). In this state, dynamin can form helices independently of the membrane. Moreover, the BSE position relative to the GTPase domain rotates by 90°, showing that GTP hydrolysis induces a conformational change compatible with constriction (*Chappie et al., 2010*; *Chappie et al., 2011*; *Sundborger et al., 2014*; *Chappie et al., 2009*). But as the structure of this transition state does not fit into the cryo-EM structure of the most constricted state of the dynamin helix (*Sundborger et al., 2014*), it is possible that the GDP·AlF$_4$-bound state of dynamin induces stresses within the helix that lead to its disassembly.

The dynamics of the fission reaction has been studied extensively in cells; it requires 5–10 s and is highly stochastic (*Merrifield et al., 2005*), consistent with in vitro fission rates in optimal conditions (*Morlot et al., 2012*). The minimal number of dynamin monomers for fission reaction is between 26 and 28, suggesting that at least one complete helical turn of dynamin is required for fission (*Cocucci et al., 2014*; *Grassart et al., 2014*). However, when GTP is hydrolyzed relative to fission in vivo is not known as in cells measuring or following the activity of a specific enzyme remains a challenge. Thus, how dynamin's GTP hydrolysis is coupled to constriction or to disassembly in vivo has not been addressed yet.

## Results

In order to follow the GTPase activity of dynamin in vivo, we ought to isolate a conformation-specific nanobody that preferentially binds one of the specific nucleotidic states of dynamin. To isolate a conformation specific nanobody against GTP-loaded dynamin-2, we screened a phage-display library of synthetic nanobodies (*Moutel et al., 2016*) against recombinant human dynamin-2 loaded with GMPPCP. Further characterization of 192 selected clones, allowed us to exclude false positives (see Materials and methods, Supplementary methods and *Figure 1—figure supplement 1*), and isolate 33% of positive clones. Based on their affinities in dot-blot assays (see Supplementary methods), we chose five dynamin-binding nanobodies and tested them in pull-down assays against various nucleotidic states of dynamin. Pull-downs were performed with dynamin assembled onto SUPported bilayers with Excess membrane Reservoir (SUPER) templates (*Pucadyil and Schmid, 2008*; *Neumann et al., 2013*) adapted for pull-down assays: magnetic silica beads were coated with lipid membranes, from which dynamin-coated membrane tubules can be formed, as dynamin binding is lost when membranes are dissolved with saponin (*Figure 1—figure supplement 2*). Beads are further pulled down with a magnet. All five nanobodies bound dynamin-2 efficiently regardless of its nucleotide load. As GTP analogs substantially increased the amount of dynamin pulled down (*Figure 1—figure supplement 3*), consistent with their ability to promote dynamin oligomerization (*Carr and Hinshaw, 1997*), GMPPCP-loaded dynamin-2 was used as a reference to ensure that the amount of dynamin pulled down was the same in every nucleotide condition.

We then tested our nanobodies against dynamin-1. We found one nanobody, that we named dynab, that preferentially binds to GDP·AlF$_4$ loaded dynamin-1 (*Figure 1A*). Overall, dynab has a lower affinity for dynamin-1 than for dynamin-2 (see *Figure 1A*). Dynab does not bind dynamin one through its PRD domain (*Figure 1—figure supplement 5A*). Since the GDP·AlF$_4$ -bound state of dynamin mimics the GTP hydrolysis transition state, dynab could potentially track the GTPase activity of dynamin by binding dynamin only during GTP hydrolysis. To test this in vitro, we first pulled-down dynab with fission defective mutants K44A and K142A. K44A has a reduced affinity for nucleotides, and thus a lower overall GTPase activity (*Damke et al., 1994*), while K142A has almost normal GTPase activity, while its helix cannot constrict (*Marks et al., 2001*). The results were similar to the

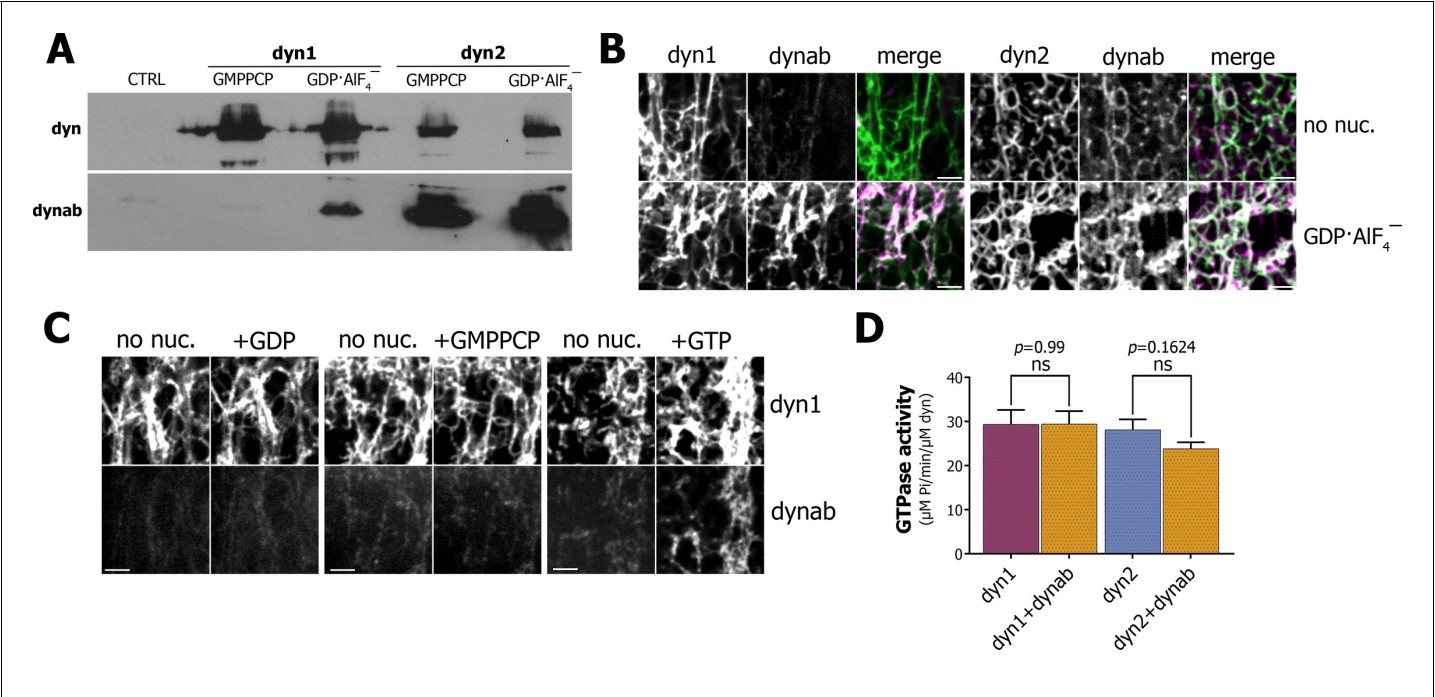

**Figure 1.** Characterization of dynab, conformation-specific nanobody. (**A**) Western blot analysis of dynamin-coated magnetic SUPER templates used as bait for pulling down of dynab. (**B**) Maximum projections of Z-stacks images depicting membrane sheets with different staining pattern of dynab on dynamin-1 and dynamin-2 tubules without nucleotide or coupled with GDP·AlF$_4$. Scale bars 3 μm. (**C**) Frames from time-lapse images of dynamin-1 tubulated membrane sheets with dynab before and after the injection of 1 μl of 1 mM GDP, GMPPCP or GTP. Scale bars 3 μm (**D**) GTPase activity of dynamin-1 and 2 with or without dynab, (bars are averages of values from five replicates ±SEM, unpaired t-test, ns = $p$ value>0.05). Source files and statistical report for panel D are available in *Figure 1—source data 1*.

DOI: https://doi.org/10.7554/eLife.25197.002

The following source data and figure supplements are available for figure 1:

**Source data 1.** (panel D) Malachite Green Assay.

DOI: https://doi.org/10.7554/eLife.25197.009

**Figure supplement 1.** Validation of dynamin-2 positive nanobodies.

DOI: https://doi.org/10.7554/eLife.25197.003

**Figure supplement 2.** SUPER templates pull-down assay control.

DOI: https://doi.org/10.7554/eLife.25197.004

**Figure supplement 3.** Dynamin binding on SUPER templates with and without nucleotides.

DOI: https://doi.org/10.7554/eLife.25197.005

**Figure supplement 4.** Verification of the knock-out procedure for dynamin triple KO cells.

DOI: https://doi.org/10.7554/eLife.25197.006

**Figure supplement 5.** Pull-downs of dynamin mutants with dynab.

DOI: https://doi.org/10.7554/eLife.25197.007

**Figure supplement 6.** Membrane sheets assay with dynamin 1-GDP·AlF$_4$ and dynab on fluorescent lipids, scale bar: 10 μm.

DOI: https://doi.org/10.7554/eLife.25197.008

wild-type (*Figure 1—figure supplement 5B*), which may indicate that even if the mutants have deficient GTPase activity, they can be locked into the same transition state when loaded with GDP·AlF$_4$.

As results of the pull-down with GTPase defective mutants were not conclusive, we used a membrane sheets assay, in which lipids are dried onto a coverslip and rehydrated to form stacks of lipid bilayers (*Roux et al., 2006*; *Itoh et al., 2005*). The addition of Alexa-488-dynamin-1 to the membrane sheets resulted in the formation of dynamin tubules, which can be visualized by confocal microscopy (*Figure 1B*). Atto-565-dynab weakly bound to dynamin-1 tubules. In contrast, in presence of GDP·AlF$_4$ we observed a strong binding of dynab to dynamin-1 tubules (*Figure 1B*). This confirmed that dynab binds preferentially to the transition state of dynamin-1, consistently with pull-down results. As GDP·AlF$_4$ can induce formation of membrane-free dynamin oligomers, we further

checked that dynab was co-localizing with dynamin 1-coated membrane tubules only in presence of GDP·AlF$_4$ (*Figure 1—figure supplement 6*). Using the same approach, we confirmed that dynab binds dynamin-2 tubules regardless of the nucleotide load (*Figure 1B*).

To test if dynab could detect dynamin-1 GTPase activity, we performed time-lapsed imaging of dynamin-1 tubules following the addition of a GTP solution. A sudden recruitment of dynab to the dynamin-1 tubules appeared after GTP addition (*Figure 1C*). In contrast, neither GMPPCP nor GDP additions led to binding of dynab (*Figure 1C*). We concluded that dynab binds to dynamin-1 tubules hydrolyzing GTP and is thus able to detect the GTPase activity of dynamin. Importantly, we further verified dynamin-1 and 2 GTPase activities were unchanged in presence of dynab (*Quan and Robinson, 2005*) (*Figure 1D*). Our results indicate that dynab has a stronger affinity for GDP·AlF$_4$ loaded dynamin-1 than for other states of dynamin-1 and thus detect the transient hydrolytic state of dynamin-1.

We then wondered if dynab may be used as an intrabody to track GTPase activity of dynamin-1 during endocytosis in living fibroblastic cells. Dynamin and dynab were expressed as fluorescently-tagged proteins in fibroblastic cells imaged by Total Internal Reflection Fluorescence (TIRF) microscopy which allow for visualization of single clathrin pits formation. To generate fibroblastic cells expressing solely dynamin-1 or dynamin-2, we used a triple conditional knockout cell line for the three dynamin genes (TKO cells) (*Park et al., 2013*). After removing the three dynamin genes (*Figure 1—figure supplement 4*), we co-expressed dynamin-1-mCherry or dynamin-2-mCherry and dynab-EGFP. In those cells, dynab co-localized with both 74 ± 4% of dynamin-1 and 88 ± 12% dynamin-2 punctae (*Figure 2B*).

We also used HeLa cells overexpressing dynamin-1 or 2 with dynab, expecting that overexpression could overcome the presence of endogenous dynamin-2. Dynab also co-localized with 80 ± 9% of the dynamin-1 and 97 ± 3% two punctae in HeLa overexpressing cells (*Figure 2B*). In HeLa cells, dynab colocalized with 76 ± 11% of clathrin-EGFP punctae (*Figure 2C*), showing that in these cells, dynab mostly follows the activity of dynamin oligomerized at clathrin-coated pits CCPs.

We then analyzed the dynamics of dynab/dynamin co-localization while being recruited at CCPs. By TIRF, dynamins show a Gaussian profile of intensity with time (*Merrifield et al., 2005*; *Merrifield et al., 2002*; *Doyon et al., 2011*; *Taylor et al., 2011*), which duration is 10–20 s. Dynab intensity mirrored the dynamin-2 intensity profile (panel 6, *Figure 3A*). This confirms that in vivo, dynab binds dynamin-2 independently of the nucleotide state. This also indicates that dynab can be used as an intrabody and that it interacts with fast kinetics with its target in living cells.

Interestingly, dynab intensity profile relative to dynamin-1 displayed a variety of patterns. We observed three categories of patterns: (i) concomitant dynamin-dynab peaks - events where dynab fluorescence intensity grew along with dynamin-1 intensity and peaked simultaneously (panel 1, *Figure 3A*); (ii) late dynab peaks - events where dynab peaked after dynamin-1 (panel 2, *Figure 3A*); and (iii) early dynab peaks - events where dynab peaked before dynamin-1 (panel 3, *Figure 3A*). We also observed double dynab peaks (panels 4 and 5, *Figure 3A*) as well as double dynamin peaks which in most cases were accompanied by dynab peaks (panel 5, *Figure 3A*).

These various profiles could indicate that dynab also detects dynamin GTPase activity in vivo. To test this, we blocked the fission activity of dynamin by incubating TKO and HeLa cells in a hypertonic media for 1'(*Morlot et al., 2012*; *Heuser and Anderson, 1989*). In both TKO and HeLa cells, dynamin-1 was blocked at clathrin-coated pits with no co-staining for dynab (*Figure 3B* and *Figure 3—figure supplement 1*) suggesting that dynab could not detect non-active, but assembled dynamin-1 in both TKO and HeLa cells. On the contrary, in cells expressing dynamin-2 and dynab, the hypertonic shock resulted in a complete co-localization of dynab and dynamin-2, which agrees with a non-conformation dependent interaction of dynab with dynamin-2 (*Figure 3B* and *Figure 3—figure supplement 1*). We further tested the ability of dynab to bind fission defective dynamin-1 K44A (*Damke et al., 1994*) and K142A mutants (*Marks et al., 2001*). We co-transfected HeLa cells with dynamin-1 K44A or dynamin-1 K142A tagged with mCherry along with dynab. Dynab did not co-localize with dynamin-1 K44A while it co-localized with dynamin-1 K142A (*Figure 3C*). These results confirmed that dynab detects the GTPase active forms of dynamin-1, as K44A is GTPase defective (*Damke et al., 1994*) while K142A is constriction defective but GTPase active (*Marks et al., 2001*).

From these controls, we concluded that dynab could detect dynamin-1 GTPase activity in fibroblastic cells. From the different binding profiles of dynab to dynamin-1 described above, we concluded that dynamin's GTPase activity in vivo is discontinuous and stochastic: it occurs in burst that

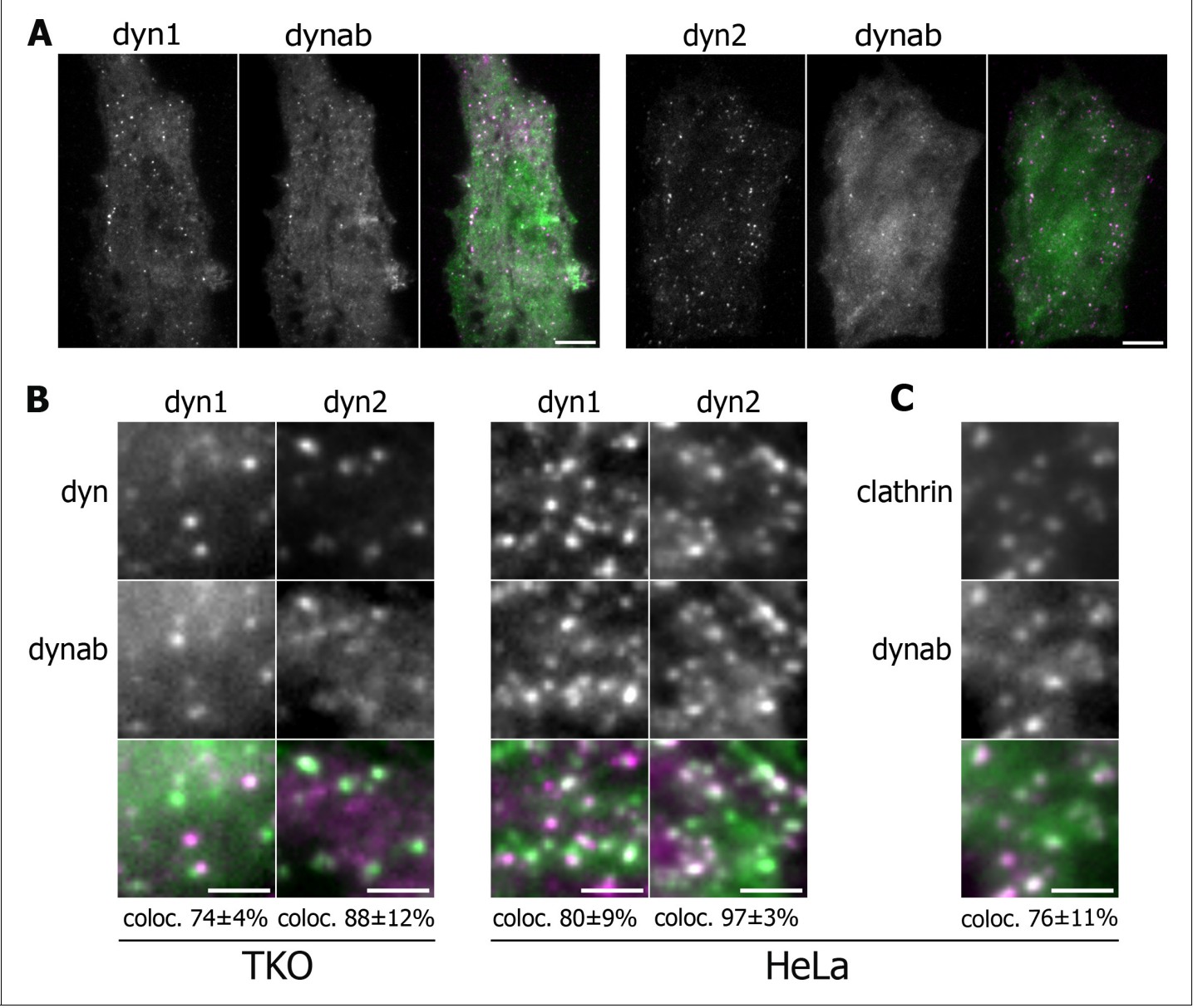

**Figure 2.** Expression in vivo of dynamin-1/2 and dynab. (A) Average images of time-lapse movies of dynamin triple knockout cells (TKO) expressing dyn1/2-mCherry and dynab-EGFP. Scale bar 10 μm. (B) Detail of TKO (left) and HeLa (right) cells expressing dyn1/2-mCherry and dynab-EGFP and relative colocalization (percentage measured over 3 cells, ±SD). Scale bar 2 μm. (C) Detail of HeLa cells expressing Clathrin-mCherry and dynab-EGFP and relative colocalization (percentage measured over 3 cells, ±SD). Scale bar 2 μm.

DOI: https://doi.org/10.7554/eLife.25197.010

are randomly distributed in time, instead of being continuous and proportional to the amount of dynamin present at the pit. We then analyzed statistically the relevance of these observations to better understand how dynamin GTP hydrolysis and polymerization are coupled.

To analyze the coupling of dynamin and dynab fluorescence profiles over time, we analyzed the time-lapse images obtained with TIRF in HeLa and TKO cells co-transfected with dynamin1- or dynamin-2 -mCherry and with dynab-EGFP. As a control, we used profiles obtained in cells transfected with both dynamin-1-EGFP and dynamin-1-mCherry as we expected them to be simultaneously recruited and disassembled from coated pits (*Cocucci et al., 2014*; *Grassart et al., 2014*; *Merrifield et al., 2002*; *Taylor et al., 2011*), called DYNctrl in the following. A software was developed to post-process all intensity profiles (see Materials and methods). Each of the profiles of either

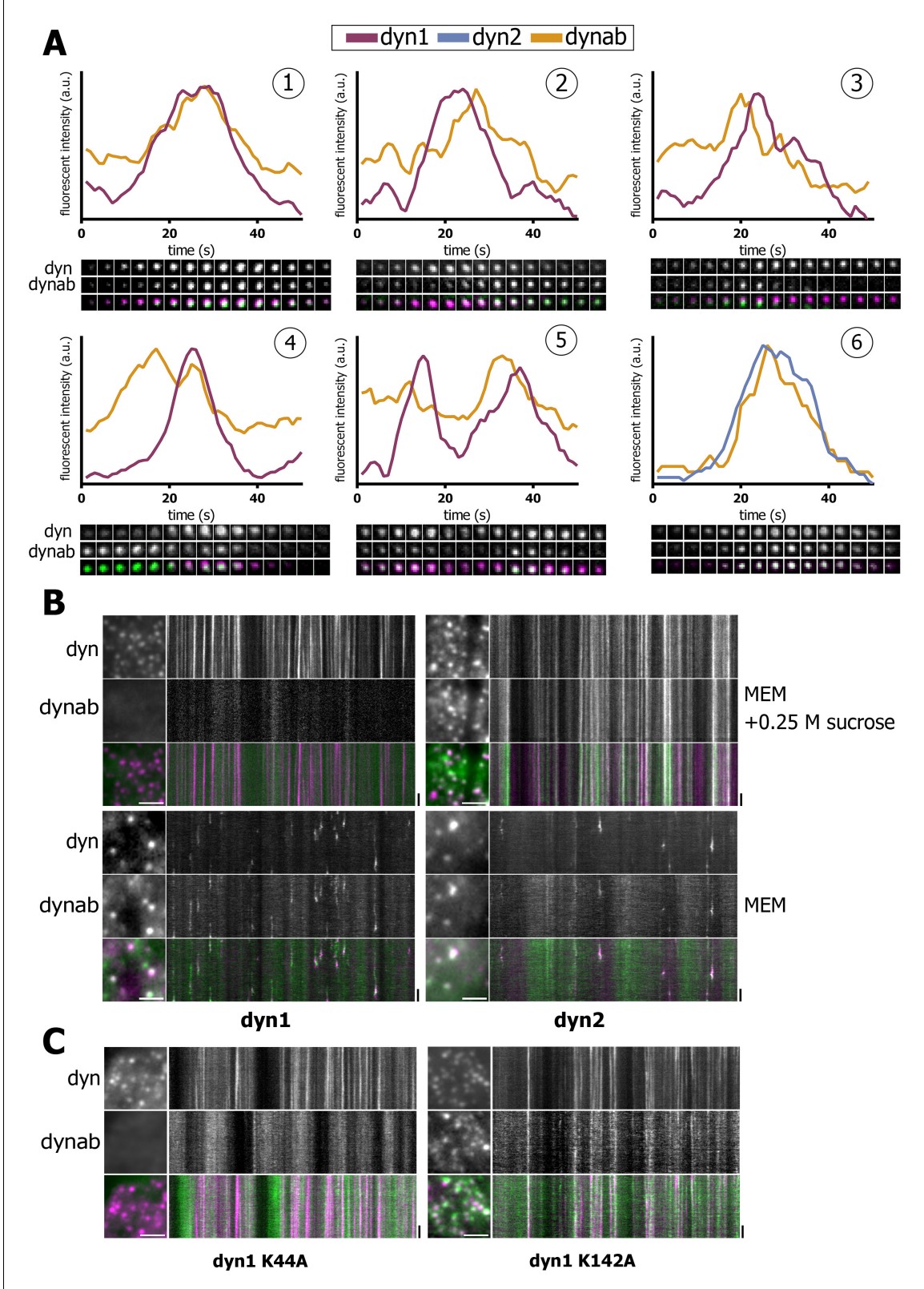

**Figure 3.** Expression in vivo of dynamin-1/2 and dynab. (**A**) Time sequences (2 s/frame) of selected dynamin-1 or 2-dynab punctae in TKO cells and relative fluorescence profiles (normalized to dynamin peak). (1): central dynab peak; (2): right dynab peak; (3): left dynab peak; (4): double dynab peak; (5): double dynamin peak (dynamin-1-dynab); (6): dynamin-2-dynab profile. (**B**) Detail and kymographs of TKO cells expressing dynamin-1–2 and dynab
*Figure 3 continued on next page*

*Figure 3 continued*

treated with 0.25 M sucrose (hypertonic shock) or in normal medium. Scale bar for insets: 3 μm, 20 s. for kymographs. (**C**) Detail and kymographs of TKO cells expressing dynamin-1 K44A or K142A and dynab. Scale bar for insets: 3 μm, 20 s. for kymographs.

DOI: https://doi.org/10.7554/eLife.25197.011

The following figure supplement is available for figure 3:

**Figure supplement 1.** Detail and kymographs of HeLa cells expressing dynamin-1/2 and dynab treated with 0.25 M sucrose (hypertonic shock).

DOI: https://doi.org/10.7554/eLife.25197.012

dynab or dynamin was fitted with 4 Gaussians. For each peak, we obtained 3 values of the Gaussian fit: the width, multiplied by four, is an estimate of the duration $d$ of the event, the peak (maximum) value $p$, and the time $t$ of the peak.

In TKO cells, the duration of dynamin-1 or 2 events was not affected by dynab, while in HeLa cells, dynab slightly reduced the duration of dynamin-1 events and had no effect on dynamin-2 (*Figure 4A*). Neither it did change the duration of clathrin-coated pits (*Figure 4—figure supplement 1*). We therefore concluded that the expression of dynab did not cause significant impairment of clathrin-mediated endocytosis.

We thus concluded that dynab did not significantly interfered with the dynamics of endocytic pits. In TKO and HeLa, dynab average duration was 70–80% of dynamin-1's average duration (*Figure 4B*). This shows that dynamin-1 is not permanently active when oligomerized at the neck of endocytic pits. As expected, dynab and dynamin-2 durations are equal in HeLa cells, while in TKO cells, a small but significant difference between dynamin-2 and dynab durations was observed (*Figure 4B*), which may indicate a slight conformation-preference of dynab for dynamin-2 in those cells.

We then analyzed how the peak of dynamin-1 GTPase activity (dynab's peak time) correlated with the dynamin-1 peak in time. We compared the distribution of time differences $\Delta t$ between dynab and dynamin-1 peaks to the ones of dynab and dynamin-2 and DYNctrl. In both TKO and HeLa cells transfected with dynamin-1, we observed that the dynab-dynamin-1 $\Delta t$ distribution is more spread than dynab-dynamin-2 or DYNctrl (*Figure 4C*). The comparison of $\Delta t$ distributions in TKO highlight that about 47% of the dynab peaks occur before dynamin-1's peak (negative values of $\Delta t$), while for dynamin-2, less than 35% of the dynab peaks occurred before. For DYNctrl, the distribution of $\Delta t$ is very narrow, and only 25% of the $\Delta t$ are negative. For HeLa cells, 46% of events have a negative $\Delta t$ for dyn1-dynab, 44% for dynamin-2-dynab and 33% for DYNctrl.

Since the duration of dynamin presence at CCPs is very variable, we calculated the time difference $\Delta t$ between peaks of dynamin and the peaks dynab, and normalized it by the duration of the dynamin peak: we obtained $\Delta t$ expressed in percentage of dynamin duration and binned the values in 5% clusters (*Figure 4C*). In TKO and HeLa cells, the distribution of $\Delta t$ for dynamin-2-dynab resembles the distribution of DYNctrl, where no time difference is expected. In contrast, the $\Delta t$ distribution between dynab and dynamin-1 revealed a more spread distribution, suggesting that the kinetics of dynamin-1 recruitment and release is not fully coupled with its hydrolysis cycle. Uncoupling is clearer considering basic statistics (*Figure 4D*): in TKO cells, if $\Delta t$ values less than ±10% are taken as simultaneous events, the majority of dynab peaks happened before or after the dynamin peak (29% peaks before and 24% of peaks after), while 47% of the events are simultaneous. Note that these 47% are not the same population than the 47% of negative $\Delta t$ values described above. For dynab-dynamin-2 and the DYNctrl, the majority of peaks (>60%) are simultaneous. In HeLa, the same statistics also show a complete uncoupling of dynab and dynamin-1 peak, with 33% of dynab peaks occurring before and 32% after dynamin-1 peak, while only 35% are simultaneous. For dynab-dynamin-2 and DYNctrl, the majority (>50%) of events are simultaneous.

Dynab also displays a double peak behavior (*Figure 3A*). In TKO cells, 8% of dynamin-1 peaks are coupled with two dynab peaks, while it represented only 5.3% and 1.6% of dynamin-2 and DYNctrl respectively. In HeLa cells, 5.8% of dynamin-1 events are coupled to double dynab peaks, 2.8% of dynamin-2 events and 2.5% of DYNctrl events. In most cases, the first peak occurs before the dynamin-1 peak, the second after (*Figure 4—figure supplement 2*). This supports the possibility that in double-peak events, the first burst of GTPase activity may happen too early, and that dynamin is not

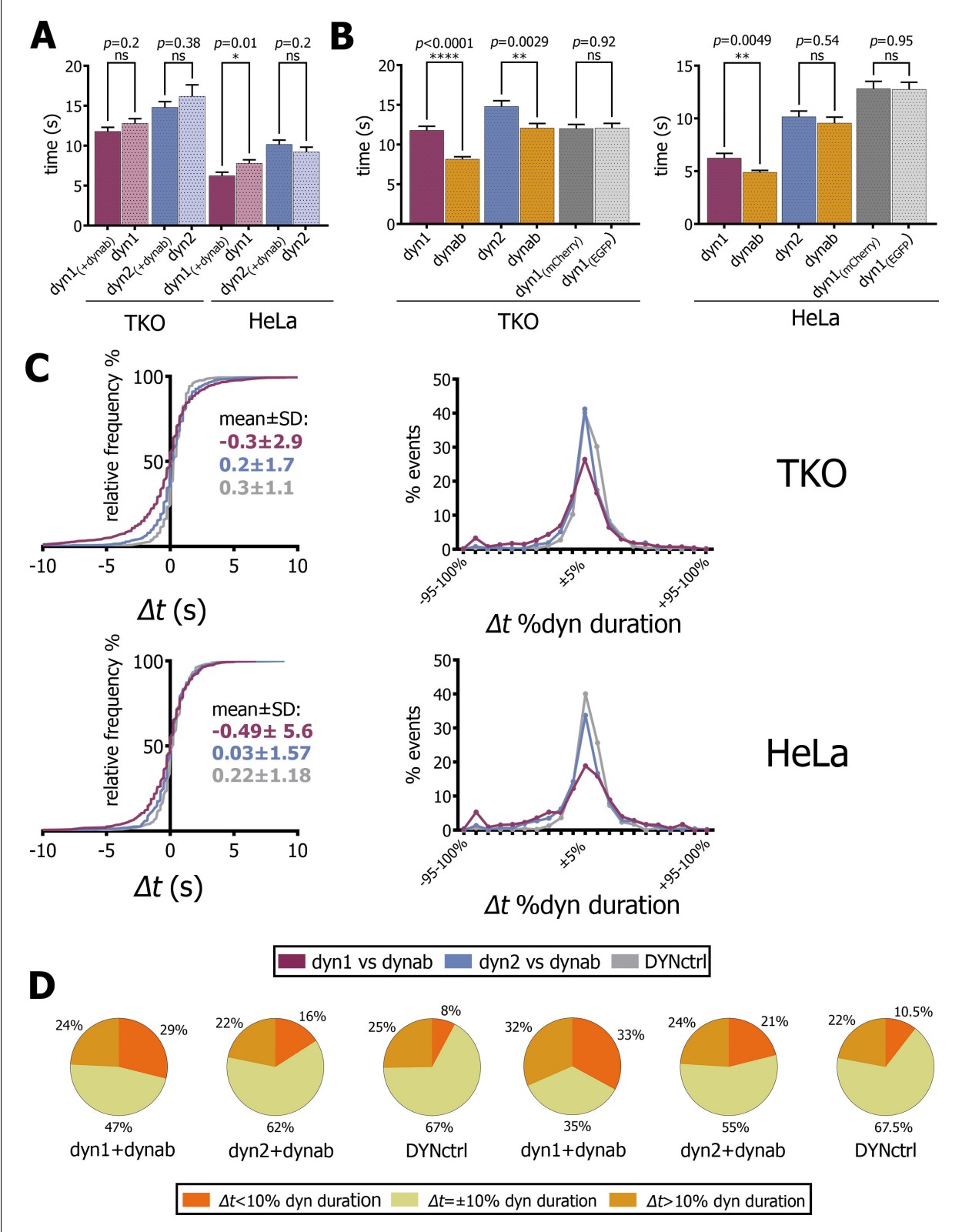

**Figure 4.** Analysis of fitted profiles. (**A**) Plot of average of dynamin-1/2 event duration with or without dynab in TKO and HeLa cells (bars are averages ± SEM, unpaired t-test, *=p value<0.05, ns = p value>0.05). (**B**) Plots of average of dynab duration compared to dynamin-1/2 duration in TKO and HeLa cells. (bars are averages ± SEM, unpaired t-test, ****=p value<0.0001, **=p value<0.05 ns = p value>0.05). (**C**) From the left: relative frequency plot of Δt values; Δt distribution expressed in percentage of relative dynamin duration; the central 0 ± 10% comprises all the values of

*Figure 4 continued on next page*

*Figure 4 continued*

the +5% and −5% cluster. Values over 100% not represented (D) Circle charts based on Δt distributions, Δt=±10% category corresponds to events for which −10% ≤ Δt ≤+10%. Total events analyzed for statistics shown in this figure: in TKO cells, dyn1-dynab = 1094 (17 cells); dyn2-dynab = 854 (9 cells); DYNctrl = 662 (9 cells); dyn1 = 833 (2 cells); dyn2 = 234 (1 cell). In HeLa cells, dyn1-dynab = 511 (13 cells); dyn2-dynab = 512 (7 cells); DYNctrl = 664 (5 cells); dyn1 = 476 (5 cells); dyn2 = 534 (5 cells). Source files and statistical report for panel A-B-C and Figure 4—figure supplements 2–4 are available in *Figure 4—source data 1*, *Figure 4—source data 2*, *Figure 4—source data 3*, *Figure 4—source data 4*, *Figure 4—source data 5*, *Figure 4—source data 6*, *Figure 4—source data 7* and *Figure 4—source data 8*.

DOI: https://doi.org/10.7554/eLife.25197.013

The following source data and figure supplements are available for figure 4:

**Source data 1.** (panel A) Comparison of duration of dynamin 1–2 events with and without dynab (expressed in seconds), and statistical report.
DOI: https://doi.org/10.7554/eLife.25197.017
**Source data 2.** (panel B) Comparison of duration of dynamin 1–2 events with dynab events in TKO cells (expressed in seconds), and statistical report.
DOI: https://doi.org/10.7554/eLife.25197.018
**Source data 3.** (panel B) Comparison of duration of dynamin 1–2 events with dynab events in Hela cells (expressed in seconds), and statistical report.
DOI: https://doi.org/10.7554/eLife.25197.019
**Source data 4.** (panel C) Cumulative probability of time difference between t(dynab)-t(dyn1-2) and DYNctrl in TKO cells, and statistical report.
DOI: https://doi.org/10.7554/eLife.25197.020
**Source data 5.** (panel C) Cumulative probability of time difference between t(dynab)-t(dyn1-2) and DYNctrl in HeLa cells, and statistical report.
DOI: https://doi.org/10.7554/eLife.25197.021
**Source data 6.** (panel C) Relative frequency Δt values; Δt distribution expressed in percentage of relative dynamin duration.
DOI: https://doi.org/10.7554/eLife.25197.022
**Source data 7.** (panel D and F) Maximum intensity values of dynab peaks vs dynamin peaks.
DOI: https://doi.org/10.7554/eLife.25197.023
**Source data 8.** Data and statistical analysis of CCPs persistence in presence of dynab in HeLa cells.
DOI: https://doi.org/10.7554/eLife.25197.024
**Figure supplement 1.** Lifetime of CCPs with or without expression of dynab in HeLa cells.
DOI: https://doi.org/10.7554/eLife.25197.014
**Figure supplement 2.** Double dynact peak distribution towards dynamin peak.
DOI: https://doi.org/10.7554/eLife.25197.015
**Figure supplement 3.** plots of maximum intensity of dynab with max intensity of dynamin.
DOI: https://doi.org/10.7554/eLife.25197.016

polymerized enough to perform its function (less than one turn), and that the second peak is the efficient activity peak that causes fission.

Considering the fact that TKO and HeLa cells gave very similar results, we concluded that overexpression of dynamin-1 in standard cell lines allows for the use of dynab without the necessity of removing endogenous dynamins. Our results indicate that the GTPase activity of dynamin occurs in bursts during polymerization of dynamin at the neck of endocytic pits, and that these bursts are not temporally correlated with the peak of dynamin presence at the pits.

We next tested if the intensity of the GTPase activity was correlated to the amount of dynamin polymerized around the neck. We found a strong correlation between the amount of dynamin-1 and dynab (*Figure 4—figure supplement 3*) present at endocytic pits. Although temporally uncoupled, it seemed that the GTPase activity is proportional to the amount of dynamin present at the pit. This supports a concerted hydrolysis of GTP by all dynamins at the neck of endocytic pits, rather than a fixed number of turns of the dynamin helix being active while other turns being inactive. In the control experiments (DYNctrl) where dynamin-1-EGFP and dynamin-1-mCherry are co-expressed, low correlation was found, whereas a strong correlation is found with dynamin-2, as expected from the high affinity of dynab for dynamin-2.

In order to check if dynab could interfere with clathrin-mediated cargo uptake, we assayed transferrin and EGF internalization in HeLa cells while overexpressing dynamin-1 or −2 in presence or absence of dynab. Although overexpression of either dynamin-1 and −2 slightly lowers transferrin and EGF endocytosis, we observed that the expression of dynab alone or together with dynamin-2 dramatically impairs transferrin internalization (*Figure 5*), while it has a minor effect on clathrin-mediated EGF internalization (low concentration of EGF are internalized mostly by CME, (*Figure 6*) (*Sigismund et al., 2008*). When dynab is co-expressed with dynamin-1, transferrin internalization is similar to internalization of transferrin in cells overexpressing dynamin-1 or dynamin-2 alone

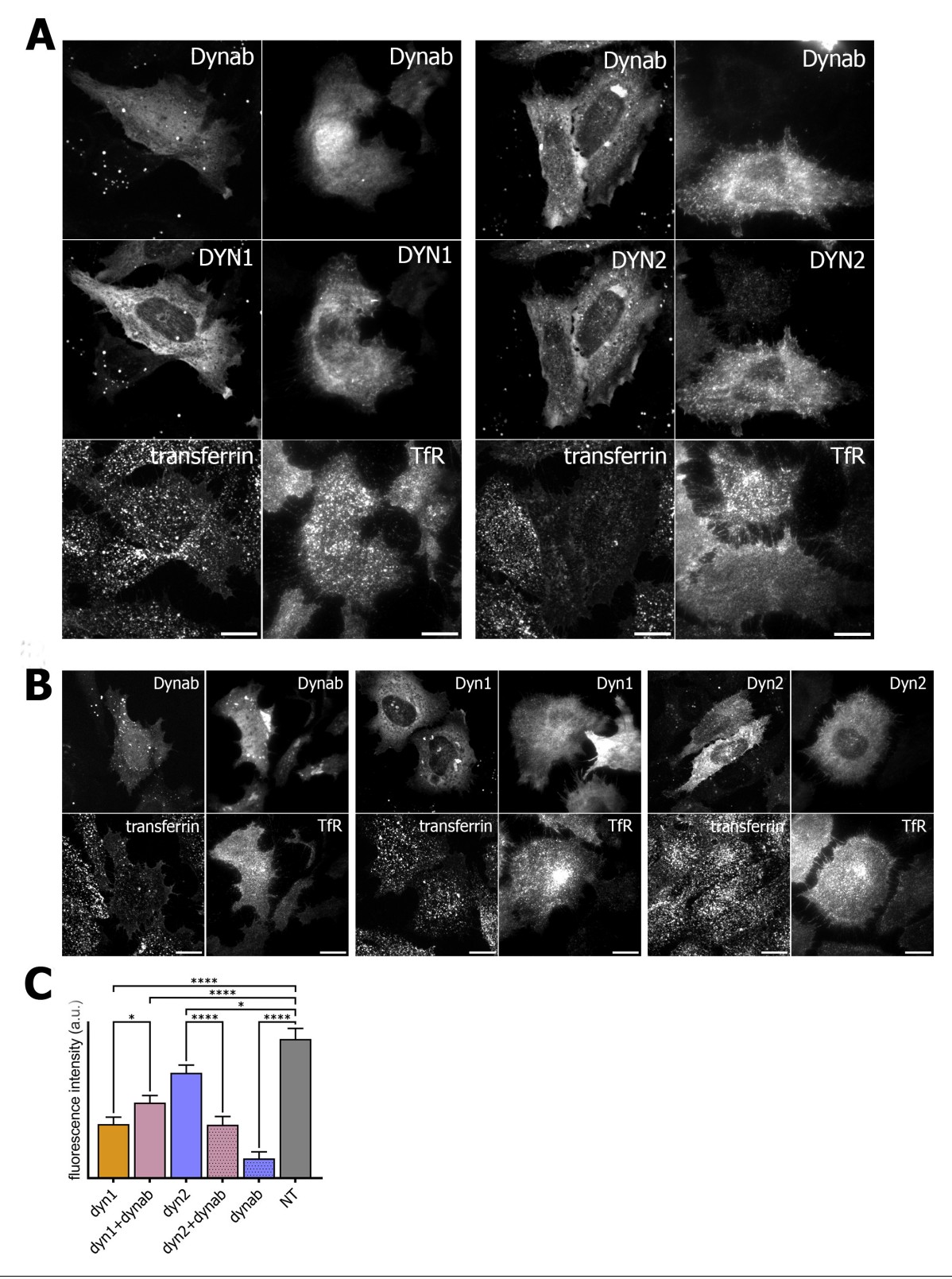

**Figure 5.** Transferrin internalization assay and TfR immunofluorescence on HeLa cells. (A) Transferrin internalization assay and TfR immunofluorescence on HeLa cells overexpressing dynamin1/2 and dynab (B) Transferrin internalization assay and TfR immunofluorescence on HeLa cells overexpressing dynamin1/2 or dynab alone. (C) Measurements (fluorescence intensity of manually fitted cell profiles) of transferrin internalization of HeLa cells expressing dynamin1/2 with dynab or dynamin 1/2 or dynab alone or not transfected (NT) (bars are averages ± SEM, unpaired t-test, *=p value<0.05,
*Figure 5 continued on next page*

*Figure 5 continued*

ns = p value>0.05). Number of cells analyzed: dyn1 n = 17; dyn1 +dynab n = 23; dyn2 n = 21; dyn2 +dynab n = 26; dynab n = 9; not transfected (NT) n = 13. Source file for panel C and *Figure 4—figure supplement 1*: *Figure 5—source data 1*, *Figure 4—source data 8*.

DOI: https://doi.org/10.7554/eLife.25197.025

The following source data is available for figure 5:

**Source data 1.** (panel C) Data and statistical analysis of transferrin internalization in HeLa cells.

DOI: https://doi.org/10.7554/eLife.25197.026

(*Figure 5*). Thus, dynab does not interfere with dynamin-1 mediated endocytosis. We however wondered how dynab could specifically interfere with the transferrin uptake mediated by dynamin-2, and not dynamin-1. As discussed above, the overall higher affinity of dynab for dynamin-2 may cause dynab to bind stronger to dynamin, potentially interfering sterically with the required interaction of the transferrin receptor (TfR) with the clathrin coat.

We thus looked the localization of the TfR in cells overexpressing dynab. In these cells, whether dynamin two was co-expressed with dynab or not, dynab expression specifically blocked the TfR at the membrane, as no endosomal pool of the TfR was visible (*Figure 5*). The endosomal pool was still visible when dynab is co-expressed with dynamin-1. Moreover, EGF receptor staining was not altered by dynab or dynamin-2 +dynab expression (*Figure 6*).

## Discussion

In this study, we visualized dynamin's GTPase activity in cells with the conformation specific nanobody dynab which interacts more strongly with the GDP·AlF$_4$ bound form of dynamin-1. To our knowledge, dynab is a unique conformation-specific nanobody in that it binds preferentially the intermediate hydrolytic state of an enzyme, allowing to follow in real time the enzymatic activity of dynamin in vivo. By using dynab in live cell imaging experiments, we showed that dynamin's GTP hydrolysis occurs in bursts and is not time-correlated with dynamin assembly.

There is a large consensus that dynamin-mediated membrane fission reaction requires the energy of GTP hydrolysis (*Antonny et al., 2016*). Two models of dynamin's mechanism differ by how the GTP hydrolytic energy is spent (*Antonny et al., 2016*). In the constriction model, this energy is proposed to be used to constrict the polymer, in which GTPase domains could act as myosins, sliding adjacent helical turns, tightening the dynamin helix. In the depolymerization model, dynamin is proposed to assemble into an already tightened helix. Fission occurs when depolymerization is triggered by GTP hydrolysis, which energy would be used for breaking non-covalent bonds between monomers. Our finding that GTPase activity of dynamin is not correlated with dynamin polymer disassembly challenges the second model.

A surprising finding is that dynamin GTPase activity occurs in burst, which suggest that it is strongly regulated. This is consistent with the requirement of a coordinated GTPase hydrolysis for efficient fission (*Liu et al., 2013*). However, assembly triggers immediate increase in dynamin GTPase activity in vitro (*Marks et al., 2001*; *Warnock et al., 1996*), which should lead to a perfect co-localization of dynab with dynamin-1 over time. Our finding of bursts of GTP hydrolysis uncoupled with the assembly/disassembly of dynamin suggests that in cells, a molecular trigger is required to allow for GTPase activation. It is even more striking that in approx. 30% of the cases, the burst of GTPase activity occurs while dynamin starts depolymerizing, suggesting that the molecular regulation of the GTPase activity is fully uncoupled from the assembly process.

Although the nature of this trigger is unknown, we can envisage several hypotheses. Various BAR (Bin-Amphiphysin-Rvs) proteins have been proposed to modulate dynamin's GTPase and fission activities (*Farsad et al., 2001*; *Yoshida et al., 2004*; *Neumann and Schmid, 2013*). One of these BAR proteins, syndapin, sees its interaction with dynamin blocked when dynamin is phosphorylated (*Anggono et al., 2006*). Furthermore, bursts of dynamin dephosphorylation occur at the synapse during active phases, suggesting that the activation of endocytosis could be triggered by dynamin-syndapin interactions (*Armbruster et al., 2013*). Finally, another trigger may come from the specific activation of dynamin GTPase activity provided by fueling of GTP by the nucleotide diphosphate kinase NM23 that binds to dynamin (*Boissan et al., 2014*). How the NM23-Dynamin interaction is

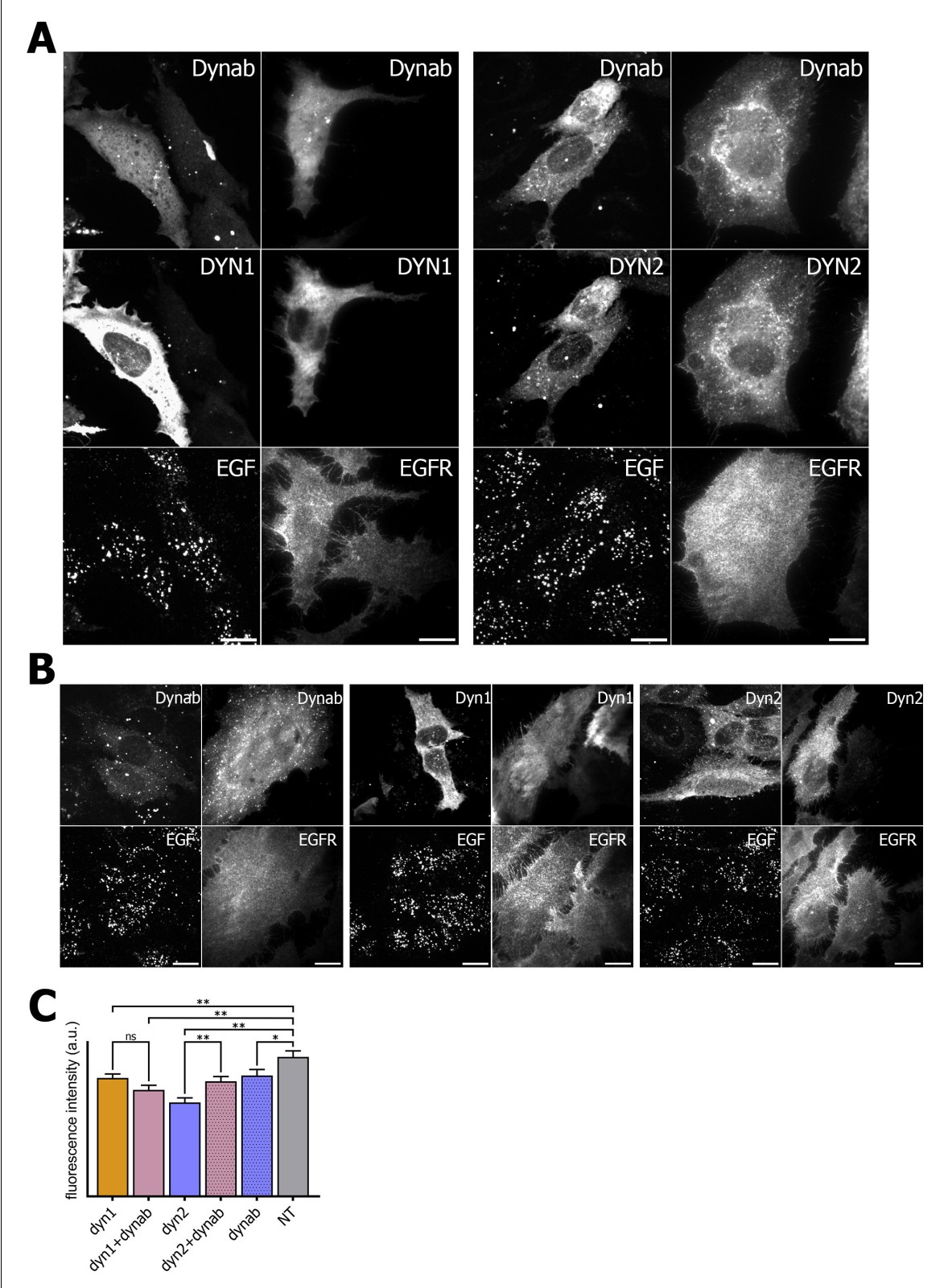

**Figure 6.** EGF internalization assay and EGFR immunofluorescence on HeLa cells. (**A**) EGF internalization assay and EGFR immunofluorescence on HeLa cells overexpressing dynamin1/2 and dynab (**B**) EGF internalization assay and EGFR immunofluorescence on HeLa cells overexpressing dynamin1/2 or dynab alone. (**C**) Measurements (fluorescence intensity of manually fitted cell profiles) of EGF internalization of HeLa cells expressing dynamin1/2 with dynab or dynamin 1/2 or dynab alone or not transfected (NT) (bars are averages ± SEM, unpaired t-test, *=p value<0.05, ns = p value>0.05).
*Figure 6 continued on next page*

*Figure 6 continued*

Number of cells analyzed: dyn1 n = 29; dyn1 + dynab n = 28; dyn2 n = 24; dyn2 + dynab n = 29; dynab n = 30; not transfected (NT) n = 25. Source file for panel C: *Figure 6—source data 1*.

DOI: https://doi.org/10.7554/eLife.25197.027

The following source data is available for figure 6:

**Source data 1.** (panel C) Data and statistical analysis of EGF internalization in HeLa cells.

DOI: https://doi.org/10.7554/eLife.25197.028

regulated is still unknown, but these three triggers may all participate in the control of dynamin GTPase activity.

## Materials and methods

### Phage display

The phage display screening was performed using a library of the recombinant VHH library NaLi-H1 (21). We used the procedure previously described in (*Dimitrov et al., 2008*; *Nizak et al., 2003*). Briefly, baculovirus purified human dynamin-2 was biotinylated using a ratio of 5:1 biotin:dynamin. This ratio was selected to be the one where substantial labeling was achieved, without interfering much with the tubulation activity of dynamin-2, as monitored by membrane sheets assay (see below). Dynamin-2 was then mixed with 1 mM guanosine-5'-[(β,γ)-methyleno]triphosphate (GMPPCP, Jena Bioscience, Jena, Germany) and subsequently with magnetic streptavidin-Dynabeads (Thermo Fisher Scientific, Waltham, MA, USA). These dynamin-2-coated beads where used as bait during the screening procedure.

A total of three sequential rounds of selection were performed by incubating the phage library with the loaded beads for 160′ in 1.4 ml PBS 0.2% Tween20 4% milk. After a total of ten washes with 10 ml of PBS 0.1% Tween20, carried out by immobilizing the beads with a magnet and removing the solution, the dynamin-bound phages were eluted from the beads with 1 ml triethylamine 100 mM.

The recovered phages were used to infect a culture of TG1 bacteria for 30′. Infected bacteria were then centrifuged to recover the pellet and plated on 2xTY agarose in order to be recovered the day after and stored or used for the following rounds of selection.

### Characterization of antibodies

192 individual clones resulted from the third round of selection were produced from TG1 bacteria, which secrete a specific MYC-tagged nanobody clone by an overnight induction with 1 mM IPTG, and analyzed by immunofluorescence and dot blot to assess their positive binding to dynamin-2.

### Immunofluorescence

HeLa MZ cells (kind gift of Dr. Lukas Pelkmans, ETHZ, Zurich [*Pelkmans et al., 2005*]) cultured on round coverslips at 70% confluency were transfected with human dynamin2 GFP PCNA plasmid (1 µg/µl, kind gift of Pietro De Camilli) using $CaCl_2$. The next day, cells were fixed with 4% paraformaldehyde for 15′. The reaction was blocked with $NH_4Cl$ 50 mM in PBS for 10′. Cells were permeabilized with 0.1% Triton, 3% BSA in PBS for 10′. Then, the coverslips were incubated onto 50 µl droplets of TG1 bacteria supernatant, containing one nanobody clone. The antibodies were detected with a primary monoclonal anti-c-myc clone 9e10 antibody (Sigma, St. Louis, MO, USA) and a secondary antibody Cy3 Donkey anti-Mouse IgG (Jackson Immunoresearch Europe, Suffolk, UK). Coverslips were then mounted on slides with Mowiol (Sigma, St. Louis, MO, USA) and screened under an epi-fluorescence microscope (Axio Z1, Zeiss, Oberkochen, Germany).

### Dot blot

A nitrocellulose membrane was mounted in a dot blot apparatus (Bio-Dot #1706545, BIORAD, Hercules, CA, USA). 200 µl of a solution of 10 nM Dynamin-2 in GTPase buffer (20 mM Hepes pH 7.4, 100 mM NaCl, 1 mM $MgCl_2$) was pipetted into each well with or without 10 µM guanosine-5'-[(β,γ)-

methyleno]triphosphate (GMPPCP, Jena Bioscience, Jena, Germany) except the last four columns that were incubated with 200 µl GTPase buffer to be used as 'no dynamin' control. After an incubation of 30', the dynamin solution was aspirated through the membrane to ensure maximal blotting of dynamin. After a blocking step with GTPase buffer milk 3%, each well was filled with 200 µl of bacterial supernatant coming from a single nanobody clone, and incubated for 30' before being removed from the wells without aspiration. The nitrocellulose was freed from the apparatus and the spots revealed by using a primary monoclonal anti-c-myc clone 9e10 antibody and an anti-mouse HRP coupled secondary antibody (both from Sigma, St. Louis, MO, USA).

## Protein purification and labelling

To purify dynamins, human dynamin-1 and human dynamin-2 were cloned without tags into pVL1392, and recombinant viruses were generated by transfecting Sf9 cells following the standard procedure of the BD baculogold expressing system (BD Bioscience, Franklin lakes, NJ USA). These viruses were then used for protein production. Infected Sf9 cells were harvested after 2.5 days of infection, and resuspended in 20 mM Hepes pH 7.4, 100 mM NaCl, 1 mM EGTA, 1 mM DTT, 1% Triton X-100 and protease inhibitor complete ULTRA tablets (Roche Applied Science, Indianapolis, IN, USA), homogenized and then centrifuged to recover the supernatant which was incubated for 2 hr. with glutathione beads coupled to GST-SH3 domain of rat Amphiphysin 1. The SH3 domain pulls down dynamin molecules through its interaction with the PRD domain of dynamins. After washing the beads with 20 mM Hepes pH 7.4, 100 mM NaCl, 1 mM EGTA, 1 mM DTT, elution was performed with 20 mM Hepes pH 7.4, 1.2 M NaCl, 1 mM $MgCl_2$. After dialysis with 20 mM Hepes pH 7.4, 100 mM NaCl, 1 mM $MgCl_2$, purified proteins were aliquoted and flash frozen for storage at −80°C. Dynamins were labelled with Atto 488 iodoacetamide (Atto-tec, Siegen, Germany), following manufacturer instructions.

To purify His-tag nanobodies from periplasm, an overnight 50 ml culture of TG1 bacteria containing a single screened clone was diluted in 500 ml 2xTY medium. The production and secretion in the periplasm of antibodies was induced with 1 mM IPTG for 4 hr at 28°C when $OD_{600}$ reached 0.5. The bacterial culture was then centrifuged and the pellet resuspended into 10 ml of 0.2 M Tris pH 8, 0.5 M EDTA, 0.5 M sucrose before being frozen in liquid $N_2$ and thawed on ice. An osmotic shock was performed on the cells by adding 40 ml of diluted buffer (50 mM Tris pH 8, 125 mM EDTA, 125 mM sucrose) and protease inhibitor (cOmplete, Roche Applied Science, Indianapolis, IN, USA) to the mix and incubating for 30' on ice. The mix was then centrifuged at 18000 g for 20' and the His tag nanobodies were purified with 250 mM NaCl, imidazole 20 mM in a Histrap FF column (GE Healthcare Life Sciences, Marlborough, MA, USA). The nanobodies were dialyzed against Tris 50 mM pH 8, NaCl 250 mM, glycerol 10%, aliquoted, flash frozen and kept at −80°C. The nanobodies were labelled with Atto NH3-ester 565 (Atto-tec, Siegen, Germany) following the manufacturer's indications.

## Pull down assays with SUPER templates

SUPER templates were prepared as described previously (*Neumann et al., 2013*) using 1.5 µm Ø magnetic silica beads (Corpuscular Inc, Cold Spring, NY, USA) and small unilamellar vesicles (SUVs) prepared as follow: a mix of 1,2-dioleoyl-*sn*-glycero-3-phosphocholine (DOPC), 1,2-dioleoyl-*sn*-glycero-3-phospho-L-serine (DOPS) and bovine brain phosphoinositide-(4,5)bisphosphate (Brain-$PIP_2$) (80:15:5% mol/mol) in chloroform for a total lipid concentration of 1 mM was dried under $N_2$ flow and resuspended in water. Vesicles were generated by performing three cycles of freezing/thawing with liquid $N_2$. To form SUVs the mix was then passed through a mini-extruder using a polycarbonated 100 nm membrane (lipids, mini-extruder and membranes were purchased from Avanti Polar Lipids, Alabaster, AL, USA). For the formation of a solution of 200 µl of SUPER templates, 0.2 mg/ml liposomes were mixed with 1 M NaCl, $5 \times 10^7$ magnetic silica beads, q.s.p. 200 µl with bi-distilled water. After incubation for 30' at room temperature, a total of three washes with bidistilled water, slow vortexing and centrifugation were performed to eliminate unbound liposomes.

For the pull-down experiment, 30 µl of SUPER templates solution were incubated for 30' with 1.2 µM of dynamin in Hepes 20 mM, NaCl 100 mM, $MgCl_2$ 1 mM (Sigma, St. Louis, MO, USA) containing 1 mM of GMPPCP or GDP·$AlF_4$. GDP·$AlF_4$ was prepared with 1 mM GDP, 10 mM NaF and 300 µM $AlCl_3$ in TRIS 20 mM pH 7.4 (Sigma, St. Louis, MO, USA). The dynamin solution was separated from

the beads by pelleting them with a magnetic holder (Dynamag-2, Thermo Fisher Scientific, Waltham, MA, USA) and were further washed 5 times with TRIS 20 mM pH 7.4 or Hepes 20 mM, NaCl 100 mM, MgCl$_2$1 mM. They were then incubated with a 6 µM nanobody solution of TRIS 20 mM pH 7.4 or Hepes 20 mM, NaCl 100 mM, MgCl$_2$1 mM with the corresponding nucleotide. The nanobody solution was removed as described above for dynamin, and the beads were washed 10 times with TRIS 20 mM pH 7.4 or Hepes 20 mM, NaCl 100 mM, MgCl$_2$1 mM before being denatured and analyzed by Western Blot using following antibodies: primary antibody for dynamin: mouse anti dynamin (BD Bioscience, Franklin lakes, NJ USA), secondary antibody anti-mouse HRP conjugated (Sigma, St. Louis, MO, USA). For nanobodies: anti-His HRP conjugated (Sigma, St. Louis, MO, USA).

## Membrane sheets

Membrane sheets were prepared as described earlier (*Itoh et al., 2005*). Briefly, 1.5 µl of a 5 mg/ml lipid solution in chloroform made of 95% Brain Polar Lipid extract (BPL) and Brain 5% phosphoinositide-(4,5)bisphosphate (Brain-PIP$_2$, Avanti Polar Lipids, Alabaster, AL, USA) was deposited on a coverslip and dried under vacuum (less than 100 µThorr) for 1 hr. In some experiments, membranes were labelled by adding 0.1% of DOPE-Atto647N (Atto-tec, Siegen, Germany). The coverslip was then mounted onto a slide using double-sided tape as spacers, forming a flow chamber of approx. 15 µl.

5 µl of a 7 µM Atto-488 dynamin solution in GTPase buffer (20 mM Hepes pH 7.4, 100 mM NaCl, 1 mM MgCl$_2$) was injected in the chamber and incubated at room temperature for 5′ in order to allow for membrane tubulation. The flow chamber was then washed with 15 µl of GTPase buffer. Then, 5 µl of 20 µM Atto-565 labeled dynab was added to the chamber, which was then imaged with a spinning-disk confocal microscope (3I Inc., Denver, CO, USA).

Loading with nucleotides was realized as follows: 1 µl of a 100 mM solution of GTP, GMPPCP or GTP was added to the open side of the chamber during imaging, allowing rapid diffusion of the nucleotide within the chamber while keeping the dynab solution. For GDP·AlF$_4$: a mix solution of dynab with GDP·AlF$_4$ was prepared with 1 mM GDP, 10 mM NaF and 300 µM AlCl$_3$ (Sigma, St. Louis, MO, USA) in TRIS 20 mM and injected in the chamber.

## Malachite green GTPase assay

Malachite green assay detects the amount of inorganic phosphate in solution and is often used to assess the activity of GTPases (*Quan and Robinson, 2005*). 200 nM of dynamin were mixed with 800 nM of dynab, 5 µl SUVs (10 mg/ml, BPL:PI(4,5)P$_2$ 95:5% mol/mol) and 500 µM GTP in a total volume of 30 µl in 96 well plates. After incubation at room temperature for 45′ malachite green reagent was added to the wells, and specific activity of dynamin was determined by comparing with standards of inorganic phosphate. Five technical replicates were performed during one experiment.

## Nanobodies cloning in EGFP vector

The sequence of a selection of positive antibodies was inserted in a modified pEGFP-N3 vector by digesting the clone sequence with BbsI and NotI and the vector with NcoI and NotI. The ligation of the sequences was performed with T4 DNA ligase (New England Biolabs, Ipswich, MA, USA).

## Dynamin triple knockout cells

Dynamin TKO cells are conditional knock-outs mice fibroblasts for dynamin-2 (flanked with CRE-lock sites) and were obtained from Pietro de Camilli, Yale University, USA (*Park et al., 2013*). Dynamin-2 was removed by treating the cells with 300 nM 4-hydroxytamoxifen (OHT, Sigma, St. Louis, MO, USA) for 5 days to induce the expression of the CRE recombinase. After the 5 days of OHT treatment, cells were transfected with Effectene (Qiagen, Hilden, Germany) following provider's instructions, with 4 µg of mouse dynamin1-mCherry or dynamin-2 -mCherry vectors (kindly provided by C. Merrifield, CNRS, Gif-sur-Yvette, France) and 4 µg dynab-EGFP vector. Cells were imaged one day after transfection using TIRF microscopy (3I Inc., Denver, CO, USA and Nikon, Tokyo, Japan).

## Generation of dynamin-1 K44A and K142A mutants

Point mutations of mouse dynamin-1-mCherry described previously were obtained using the site-directed mutagenesis kit QuickChange (Agilent Technologies, Santa Clara, CA, USA) following provider's instructions.

## Transferrin and EGF internalization assay and TfR/EGFR immunofluorescence

Transferrin and EGF internalization where performed by starving cells (with medium lacking FBS) HeLa plated on glass coverslips (expressing dynamin 1/2 in presence or absence of dynab or dynab alone) for 3 hr at 4°C, and then incubated in MEM with the addition of with 10% FBS containing either 25 µg/ml fluorescently labeled transferrin (Transferrin From Human Serum, Alexa Fluor 647 Conjugate, Thermo Fisher Scientific, Waltham, MA, USA) or 2 ng/ml fluorescently labeled EGF (Alexa Fluor 647 EGF complex, Thermo Fisher Scientific, Waltham, MA, USA) for 30'. Acid wash with 25 mM sodium acetate in MEM pH 2 for 2 min was performed to remove surface bound ligands. The medium was then neutralized by addition of 25 mM Tris in MEM. Cells were then fixed with PFA 4% and mounted on slides with Mowiol. Z-stack images (step-size of 0.5 µm) were obtained and the levels of internalization were extracted from the average fluorescent intensity of manually profiled cells.

TfR and EGFR immunofluorescence was performed by fixing transfected cells (expressing dynamin 1/2 in presence or absence of dynab or dynab alone) with 4% PFA and a subsequent step of permeabilization and blocking with 0.1% Triton 3% BSA before incubating with primary antibodies for TfR or EGFR (Abcam, Cambridge, UK) and later with far red fluorescent-coupled secondary antibody (Abcam, Cambridge, UK). For the measurements of coated pit lifetime in cells expressing or not expressing dynab, HeLa cells were transfected with calcium phosphate using 3.5 µg EGFP/ Dynab-EGFP and 1.5 µg Clathrin-mCherry constructs. Live-imaging was performed the day after transfection using TIRF microscopy and data was analyzed with ImageJ; 4 independent experiments were performed counting from 1 to 5 cells/experiment.

## Microscopes and imaging

Time-lapse images were generated with an inverted microscope (Eclipse c1 from Nikon, Tokyo, Japan) using spinning disk confocal microscopy (3I Inc., Denver, CO, USA) or motorized TIRF system (Nikon) and acquired with a Roper Scientific camera (eVolve). Images were analyzed and processed with ImageJ. Magnified images were produced by scaling the original image with bilinear interpolation. Images presented are average intensities projection of time-lapse images.

Imaging experiments were performed one day after transfection in the following fashion: TKO cells expressing dyn1 + dynab (total 17 cells) were imaged during five different days; TKO cells expressing dyn2 + dynab (total 9 cells) were imaged during two different days; TKO cells DYNctrl (total 9 cells) were imaged 1 day; TKO cells expressing dyn1 (total 2 cells) were imaged during 1 day; TKO cells expressing dyn2 (total 1 cell) were imaged during 1 day; HeLa cells expressing dyn1 + dynab (total 13 cells) were imaged during two different days; HeLa cells expressing dyn2 + dynab (total 7 cells) were imaged during two different days; HeLa cells DYNctrl (total 5 cells) were imaged 1 day; HeLa cells expressing dyn1 (total 5 cells) were imaged during 1 day; HeLa cells expressing dyn2 (total 5 cells) were imaged during 1 day.

## Data analysis and fitting

The analysis of dynamin-dynab co-localization events was performed by detecting over time all punctae of dynamin and dynab in each time sequence and saving the information. To detect the individual spots at each frame we used the detection algorithm implemented in the program Utrack developed by Jaqaman et al. (*Jaqaman et al., 2008*), with which both channels belonging to dynab-EGFP and dynamin-1/2-mCherry were analyzed.

To obtain the events (defined as bright spots observed in a set of consecutive frames with restricted or no movement) a tailored Matlab code was developed (post-Utrack) that post-processed the output from the Utrack detection software. Our code classifies each individual spot in an image into an already detected candidate event or a new candidate event. All candidate events have to fulfill the following five criteria to be considered real events. First, the sequence of spots forming the

event must show spatial correspondence over time. Second, the event has to last a minimum number of frames determined by a user threshold (set to 10 frames in our sequences), but can be missing or undetected for two consecutive frames. Third, the event peak intensity has to be at least a 40% of the highest peak detected in any other event of the whole sequence. Note that the intensity is the integral of the gray level intensity in a circle centered in the spot maximum, where the background has been subtracted. Fourth, there must be a correlation higher than 0.45 between the intensity profiles of both channels. Finally, each channel has to show a goodness-of-fit with a mixture Gaussian function higher than 0.6. The software does not discard any candidate event that does not fulfill all the criteria, but labels each of them for the final user to decide which should be considered real events.

Following, the gray-intensity profiles are shifted to ensure that the minimum intensity of the profile is zero, and the data are fitted to a cubic spline function to increase the temporal resolution four-fold, that is 0,25 frames. Next each channel of the real event profiles is fitted to a four Gaussian mixture. When only a single peak is observed in a profile the fit returns the principal Gaussian together with Gaussians fit to the background noise which are subsequently discarded. From the principal Gaussians the peak of each channel is obtained and the relative delay between them obtained with an accuracy of 0.25 frames. In addition, the parameters of the Gaussians selected are stored to obtain statistics (event duration, event intensity) and to perform the curve averaging of the functions instead of the sampled data.

All events were manually validated to correct fitting errors and remove inconsistencies. Statistics and graphs were elaborated using Graphpad Prism.

## Supplementary methods
### Phage display selection of Dynamin-binding nanobodies
Based on successful attempts to isolate conformation-specific nanobodies against other GTPases – Rab6 and tubulin (*Dimitrov et al., 2008*; *Nizak et al., 2003*) we ought to isolate a conformation specific nanobody against GTP-loaded dynamin-2. A phage-display library of single-chain nanobodies derived from llama (library NaLi-H1) including the antigen-binding variable domain (*Moutel et al., 2016*), but lacking the constant regions, was screened against purified human dynamin-2 in the presence of GMPPCP. Dynamin-2 was chosen because of its ubiquitous distribution, and GMPPCP because of its stability and low dissociation constant with dynamin, allowing for the conservation of the GTP-loaded state during the screen. The screen was carried out by attaching 40 micrograms of biotinylated dynamin-2 to magnetic, streptavidin-coated beads, in the presence of 0.2 micromolar GMPPCP.

These dynamin-coated beads were used to enrich fractions with phages binding to GMPPCP-loaded dynamin-2 through three successive rounds of selections. We found a significantly increased number of selected phages through the three rounds, indicating an efficient selection. Single dynamin-binding phages were sorted out by both immunofluorescence and dot blot analysis (see *Figure 1—figure supplement 1* and Materials and methods). *Figure 1—figure supplement 1* presents results of this characterization. By immunofluorescence, antibodies binding to cells proportionally to the dynamin2-EGFP expression level were selected as positive (*Figure 1—figure supplement 1*, left panel). Specific dynamin-2 binding was confirmed by dot blot analysis, where purified human dynamin-2 loaded with various nucleotides was immobilized onto the membrane, and nanobodies used as primary antibodies (*Figure 1—figure supplement 1*, right panel). Even though differences of binding intensities were observed with various nucleotidic loads in single assays, poor reproducibility of the results indicated that dot blot analysis could not be used for isolating a conformation-specific nanobody.

Out of a total of 192 analyzed clones, we found that 33% of the clones where efficiently binding dynamin-2 in both assays. We further screened these positive clones for with conformation-specific by pull-down assays.

### Isolation of dynab, a GDP·AlF$_4^-$-conformation-specific nanobody against dynamin-1
To further screen dynamin-binding nanobodies for affinity changes with the nucleotidic states of dynamin, we adapted SUPported bilayers with Excess membrane Reservoir (SUPER) templates for

pull-down assays (*Pucadyil and Schmid, 2008*; *Neumann et al., 2013*). SUPER templates consist of silica-beads covered with lipid membranes that can be tubulated and fissioned by dynamin. As dynamin binds to SUPER templates in its functional form (helical collar around a membrane tube), and without the requirement of tagging dynamin, we performed pull-downs of 5 dynamin-binding nanobodies using magnetic SUPER templates – where silica beads were replaced by magnetic silica beads – and pelleted the beads with a magnet. Specific binding of dynamin to the lipid coating was checked by treating the beads with 0.5% saponin (*Figure 1—figure supplement 2*).

Dynamin-binding clones to be tested were selected first for their binding intensity in dot blot assays, and second for their purification yield after recombinant expression in bacteria. The fast magnetic SUPER template procedure allowed us to test each clone for binding to dynamin-2 loaded with different nucleotides. Surprisingly all the clones we tested bound dynamin-2 efficiently regardless of its nucleotide load (see *Figure 1A*). This may be because our selection procedure retained only nanobodies binding dynamin-2 with high affinity. Consistent with this idea, conformation-specific antibodies usually have a moderate affinity. Also, we noticed that GTP analogs substantially increased the amount of dynamin binding to magnetic SUPER templates (*Figure 1—figure supplement 3*), probably because they increase dynamin oligomerization (*Carr and Hinshaw, 1997*). Thus, GMPPCP-loaded dynamin-2 was used as a reference instead of nucleotide-free dynamin in the following experiments.

Despite the unsuccessful results with dynamin-2, we reasoned that some of the dynamin-binding nanobodies may have lower affinity for dynamin-1, and thus tested them against dynamin-1. We found one clone, that we named dynab, that preferentially bind to the GDP·AlF$_4^-$ loaded dynamin-1 (*Figure 1A*), but with an overall lower affinity for dynamin-1 than for dynamin-2. Interestingly, dynab specifically detected the transition state of dynamin (GDP·AlF$_4^-$ bound state), which is supposed to be transient and concomitant with the GTP hydrolysis. We concluded that dynab could potentially track the GTPase activity of dynamin.

## Acknowledgements

The authors thank Marko Kaksonen for their critical reading of the manuscript and useful discussion and Christien Merrifield and Pietro De Camilli for the materials provided. They also thank Tomas Kirchhausen for his useful insights on the project. VG and AR acknowledge funding from the Initial-Training Network TRANSPOL (Marie Curie action grant #264399). The work was supported by Human Frontier Science Program CDA-0061–08, the Swiss National Fund for Research Grant N°31003A_130520 and the European Research Council Starting Grant N° 311536 (2011 call) to AR. The work in FP lab was supported by the Institut Curie, the CNRS and by grants from the Fondation pour la Recherche Médicale (DEQ20120323723) and from the Agence Nationale pour la Recherche (ANR-12-BSV2-0003-01).

## Additional information

### Funding

| Funder | Grant reference number | Author |
| --- | --- | --- |
| Human Frontier Science Program | CDA-0061-08 | Aurélien Roux |
| Schweizerischer Nationalfonds zur Förderung der Wissenschaftlichen Forschung | Grant 31003A_130520 | Aurélien Roux |
| H2020 European Research Council | Starting Grant 311536 (2011 call) | Aurélien Roux |
| Seventh Framework Programme | Marie Curie ITN grant #264399 | Aurélien Roux |
| Agence Nationale de la Recherche | ANR-12-BSV2-0003-01 | Franck Perez |
| Centre National de la Recherche Scientifique | | Franck Perez |

Fondation pour la Recherche       DEQ20120323723            Franck Perez
Médicale

The funders had no role in study design, data collection and interpretation, or the
decision to submit the work for publication.

## Author contributions
Valentina Galli, Conceptualization, Data curation, Formal analysis, Investigation, Methodology, Writing—original draft, Writing—review and editing; Rafael Sebastian, Software, Writing—original draft; Sandrine Moutel, Resources, Validation, Investigation, Methodology; Jason Ecard, Formal analysis, Investigation, Methodology; Franck Perez, Conceptualization, Funding acquisition, Methodology, Writing—original draft, Writing—review and editing; Aurélien Roux, Conceptualization, Supervision, Funding acquisition, Writing—original draft, Project administration, Writing—review and editing

## Author ORCIDs
Franck Perez       http://orcid.org/0000-0002-9129-9401
Aurélien Roux       http://orcid.org/0000-0002-6088-0711

## Decision letter and Author response
Decision letter https://doi.org/10.7554/eLife.25197.036
Author response https://doi.org/10.7554/eLife.25197.037

# Additional files

## Supplementary files
• Source code 1. postUtrack_1.
DOI: https://doi.org/10.7554/eLife.25197.029

• Source code 2. postUtrack_2_GenerateProfiles.
DOI: https://doi.org/10.7554/eLife.25197.030

• Supplementary file 1. Supplementary File_Manual _Code.
DOI: https://doi.org/10.7554/eLife.25197.031

• Transparent reporting form
DOI: https://doi.org/10.7554/eLife.25197.032

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
