## [Decision Letter]

Thank you for submitting your article "Uncoupling of dynamin polymerization and GTPase activity revealed by the conformation-specific nanobody Dynab" for consideration by *eLife*. Your article has been favorably evaluated by Anna Akhmanova (Senior Editor) and three reviewers, including Christien Merrifield (Reviewer #1) as the Reviewing Editor. The following individual involved in review of your submission has agreed to reveal their identity: Marjin Ford (Reviewer #3).

The reviewers have discussed the reviews with one another and the Reviewing Editor has drafted this decision to help you prepare a revised submission.

Summary:

In this paper, the authors develop a nanobody (Dynab) that recognizes the GTP bound state of dyn1. They show that, in cells, the recruitment of dynab is more tightly coupled to dynamin-1 recruitment than to dynamin-2 recruitment. They interpret the temporal mismatch between Dynab and dynamin-1 recruitment to indicate that dynamin GTP hydrolysis occurs stochastically and is only weakly coupled to dynamin-1 recruitment. They propose that their finding that the GTPase activity of dynamin is not correlated with dynamin polymer disassembly challenges one published model of dynamin action. Three reviewers have now read your paper and have come to similar conclusions. Overall all reviewers found the paper interesting and certainly provocative. However, some common concerns were raised that warrant further experiments, which would clarify the specificity and cellular effects of Dynab.

Essential revisions:

1) It would be important to understand better where Dynab binds to dynamin-1 and what transition state dynamin is in. This is essential information for interpreting the data. Furthermore, it is unclear why Dynab recognizes dynamin-2 irrespective of GTP bound state while it only recognizes GTP-bound dynamin-1. Dynamin-1 and dynamin-2 are ~80% homologous and it is safe to presume that the core mechanism of GTP hydrolysis / membrane scission is common to both. So knowing where Dynab binds dynamin-1 and understanding what underlies the specificity of Dynab is key to interpreting the data and will inform (i) the core mechanism of dynamin mediated scission and (ii) the peculiarities of dynamin-1 versus dynamin-2 mediated scission. Using minimal constructs and perhaps dynamin1/2 chimeras would also be useful here for in vivo work.

While extending characterization of Dynab, we would like to ask you to take into account the considerations listed below and, where possible and not too time-consuming, address them.

a) Considering that Dynab recognizes the GDP-AlF bound state of dynamin-1, Dynab signals observed in the experiments could be of dynab binding to non-lipid bound soluble dynamin oligomers rather than to the helical collars that mediate membrane fission. The authors might consider fluorescently labeling the lipid in the in vitro assays, which will confirm if dynamin is still bound to the lipid in the presence of GDP.ALF^-^. Alternatively, negative stain electron microscopy with gold labeled Dynab might be employed to show if the nanobody is labeling dynamin tubes.

b) It would be nice if the authors could determine whether Dynab primarily recognizes the dynamin G domain dimer, which is known to be stabilized by dynamin binding to GDP-AlF. Preferential recognition of the dimer by the nanobody may alter the natural kinetics of dynamin when the nanobody is present.

c) In addition to dynamin bound to GTP, GMPCP and GDP-AlF_4_, the authors might consider looking at various dynamin mutants such as K44A, K142A, and ΔPRD bound to these same GTP analogues in vitro.

2) A second, related question concerns the coincidence of peak Dynab versus dynamin-1 recruitment. Dynab binds to dynamin in the crowded molecular scrum at sites of endocytosis and will presumably block or interfere with the recruitment of some of dynamins' partners – and likewise recruitment of Dynab to dynamin will be blocked by invisible players. This is important because, for any given endocytic event, this will (i) influence the amount of Dynab bound and so (ii) influence where peak Dynab recruitment falls relative to peak dynamin recruitment. A simple control could be to co-express dynamin-1-GFP and dynamin-1-mCherry in the TKO background and find how often the peaks of green/red dynamin recruitment coincide. This will give some crude insight into the 'power' of the measurement – i.e. whether it is even reasonable to presume that the fluorescence signal of Dynab and dynamin-1 will coincide given the noisy signals.

3) It would be advisable that the authors test the uptake of a model cargo (transferrin) in the presence/absence of Dynab expression.

4) There are several issues with the analysis. First, and in a related vein to (16) it is striking that in general the kinetics of dynamin recruitment are consistently slower in the presence of Dynab (Figure 4) and perhaps the pair-wise comparisons used here will mask this global trend. An ANOVA type analysis using all the data in Figure 4 would be more appropriate here. Also the metric of 'time divided by time' has no units and the authors should use a different way to express this (perhaps using fractional notation).

[Editors' note: further revisions were requested prior to acceptance, as described below.]

Thank you for resubmitting your work entitled "Uncoupling of dynamin polymerization and GTPase activity revealed by the conformation-specific nanobody Dynab" for further consideration at *eLife*. Your article has been favorably evaluated by Anna Akhmanova (Senior Editor), a Reviewing Editor and two external reviewers.

After careful consideration and consultation between the reviewers we have reached a consensus. We have all agreed that the work reported in your manuscript is important and that your finding that dynab can detect the GTP loaded state of Dnm1 is new and important, and that this will prove a useful tool for other researchers. However, we have also agreed that your new finding that dynab blocks Tfn uptake (but not EGF uptake) is a key piece of information that the readers need to have at hand when they read the manuscript. Therefore, these data should be moved from their current position in the supplementary data to a figure in the main text. Also, we must ask for one further experiment – that is to measure coated pit lifetime in control cells versus cells expressing dynab. These results should be included with the Tfn/EGF data in one figure which evaluates the effects of dynab on endocytosis.

---

## [Author Response]

Essential revisions:1) It would be important to understand better where Dynab binds to dynamin-1 and what transition state dynamin is in. This is essential information for interpreting the data. Furthermore, it is unclear why Dynab recognizes dynamin-2 irrespective of GTP bound state while it only recognizes GTP-bound dynamin-1. Dynamin-1 and dynamin-2 are ~80% homologous and it is safe to presume that the core mechanism of GTP hydrolysis / membrane scission is common to both. So knowing where Dynab binds dynamin-1 and understanding what underlies the specificity of Dynab is key to interpreting the data and will inform (i) the core mechanism of dynamin mediated scission and (ii) the peculiarities of dynamin-1 versus dynamin-2 mediated scission. Using minimal constructs and perhaps dynamin1/2 chimeras would also be useful here for in vivo work.

We agree with the reviewers that understanding how and why dynab binds Dyn2 regardless of its load as compared to Dyn1 is essential. On the recommendation of the reviewers, we have thus performed pull-downs with Dyn1-ΔPRD, and the G-G construct (G domain+BSE) crystallized in Chappie et al. Nature 2010. We also performed pull-downs with the full-length dyn1 mutants K44A and K142A. The results are:

- Pulling down dynab with Dyn1-ΔPRD on magnetic SUPER templates gives essentially the same results than full-length Dyn1, showing that, as expected, dynab does not bind the PRD domain of dynamin (see Figure 1—figure supplement 5).

- We were able to pull-down the G-G construct using dynab as a bait (see Author response image 1), as seen from the slightly more intense bands in the G-G monomer (no nucleotide) and dimer (in presence of GDP.AlF_4_^-^) columns compared to the control w/o dynab, and the S41 mutant (no dimerization, no conformational change). As these results are not striking and that changes in intensity could be linked to several common technical issues in pull-down assays, we do not think these results are of sufficient quality to be published. We think that dynab probably binds to dynamin in the helical polymer predominantly, which would explain the very weak interaction characterized by this pull-down.

- Pulling-down dynab with the full-length K44A and K142A mutants on SUPER magnetic templates in the presence of GMP-PCP or GDP.AlF_4_^-^ gave essentially the same results than the WT full-length dynamin 1 (see Figure 1—figure supplement 5). Since both mutants are not locked in only one state, as K44A has a minimal GTPase activity, and that K142A has a normal GTPase activity, we think the mutants can be put into the same state than the WT by loading them with the same non-hydrolysable GTP analog.

In short, since we observed dynab binding only when dynamin was oligomerized onto membranes, we think dynab detects the polymeric helix of dynamin, in the transition state. We think dynab could bind to two adjacent helix turns at the same time, which would mean that it binds when the turns are tightly paired together by G-G bonds in the transition state. However, it may not bind the G-G dimer only, which means it probably binds parts of the stalks that are closely apposed when the turns are paired.

In respect to the different binding of dynab to dynamin 1 and 2, the reviewers raised a very important point: since the sequence homology between the two proteins is higher than 80% and admitting that the mechanism and structure of dynamin 1 and 2 polymers are similar (although there is no crystal structure of Dyn2 and not many in vitro assays using Dyn2), dynab should bind the two proteins with the same conformation sensitivity, whereas in most of our in vitro assays, dynab exhibited binding to dynamin2 regardless of its nucleotide load.

But we also observed that dynab has an overall lower affinity for dynamin1 than dynamin2 (see Figure 1).

By highlighting the different amino acids (see Author response image 2), it is possible to observe that in the BSE, G-domain and the stalk, these differences would be sufficient to cause at least a difference of affinity of dynab for Dynamin1 as compared to Dynamin2.

**Author response image 2. respfig2:** 

Also, we think that this higher affinity of dynab for dynamin2 may mask its ability to recognize specifically the GDP.AlF_4_^-^ state in the in vitro assays. With a higher affinity, dynab will bind anyway to dynamin2, even perhaps forcing dynamin2 to adopt the conformation dynab prefers to bind to. Several of our observations are in line with this hypothesis:

1) dynab seems to weakly inhibit the GTPase activity of dynamin2 in GTPase assays (see Figure 1), which may be due to the fact that dynab partially blocks a specific conformational step in the GTPase cycle.

2) The duration of dynab peak at CCPs in cells is smaller than the one of dynamin, both for dyn1 and dyn2, whereas the controls (dyn-mCherry vs. dyn-GFP, DYNctrl) are perfectly identical. This suggest that dynab recognize dyn2 in a similar way than dyn1 in cells.

3) In further support of this, max intensities of dynab are proportional to max intensities of both dyn1 and dyn2 in TKO cells (Figure 3), suggesting that dynab could detect the same conformation in both dyn1 and dyn2.

In conclusion, we do not think that dynab is detecting dyn1 and dyn2 in a very different manner, and we even think that dynab could bind dyn2 in a conformation specific manner. However, due to its higher affinity for dyn2, this effect may be partially masked in our results.

While extending characterization of Dynab, we would like to ask you to take into account the considerations listed below and, where possible and not too time-consuming, address them.a) Considering that Dynab recognizes the GDP-AlF bound state of dynamin-1, Dynab signals observed in the experiments could be of dynab binding to non-lipid bound soluble dynamin oligomers rather than to the helical collars that mediate membrane fission. The authors might consider fluorescently labeling the lipid in the in vitro assays, which will confirm if dynamin is still bound to the lipid in the presence of GDP.ALF-. Alternatively, negative stain electron microscopy with gold labeled Dynab might be employed to show if the nanobody is labeling dynamin tubes.

It is a fair point to say that dynab could bind to the oligomers of dynamin regardless of whether they are bound to the membrane or not. This is however unlikely since in the in vitro experiments (membrane sheets assay), unbound dynamin (whether monomeric or oligomeric) was washed-out of the chamber before addition of dynab. Moreover, in previous publication, dynamin staining was colocalizing with membrane tubes (see Figure 1 in Roux et al. Nature 2006). But we still performed the control proposed by the reviewers and we now show in Figure 1—figure supplement 6 that, using fluorescent lipids, dynamin coats the membrane tubules while interacting with dynab in presence of GDP·AlF_4_‾.

b) It would be nice if the authors could determine whether Dynab primarily recognizes the dynamin G domain dimer, which is known to be stabilized by dynamin binding to GDP-AlF. Preferential recognition of the dimer by the nanobody may alter the natural kinetics of dynamin when the nanobody is present.

The reviewers here have a very good point, and we have performed GST pulldowns with the G-G construct that was published before (see Author response image 1). As said above, we were able to pull-down the G-G construct using dynab as a bait, with and without GDP.AlF_4_^-^. As described above, these results are not conclusive, and we are not able to definitively confirm that dynab binds to the dimeric G-G form. It probably means that dynab binds to the oligomeric form of dynamin, including some parts of the stalk.

*c) In addition to dynamin bound to GTP, GMPCP and GDP-AlF_4_, the authors might consider looking at various dynamin mutants such as K44A, K142A, and ΔPRD bound to these same GTP analogues* in vitro.

As suggested by the reviewers, we pulled down dynab with the full-length K44A and K142A mutants on SUPER magnetic templates in the presence of GMP-PCP or GDP.AlF_4_^-^. The results were essentially the same than the WT full-length dynamin 1 (see new Figure 1—figure supplement 5). Since both mutants are not locked in only one state, as K44A has a minimal GTPase activity, and that K142A has a normal GTPase activity, we think the mutants can be put into the same state than the WT by loading them with the same non-hydrolysable GTP analog, which explain why we have the same results with mutants than WT.

2) A second, related question concerns the coincidence of peak Dynab versus dynamin-1 recruitment. Dynab binds to dynamin in the crowded molecular scrum at sites of endocytosis and will presumably block or interfere with the recruitment of some of dynamins' partners – and likewise recruitment of Dynab to dynamin will be blocked by invisible players. This is important because, for any given endocytic event, this will (i) influence the amount of Dynab bound and so (ii) influence where peak Dynab recruitment falls relative to peak dynamin recruitment. A simple control could be to co-express dynamin-1-GFP and dynamin-1-mCherry in the TKO background and find how often the peaks of green/red dynamin recruitment coincide. This will give some crude insight into the 'power' of the measurement – i.e. whether it is even reasonable to presume that the fluorescence signal of Dynab and dynamin-1 will coincide given the noisy signals.

The point that the reviewer is making here came to our mind, and we thus used the exact same control proposed here by the reviewer. In Figure 4, all statistical analyses are actually compared to the analysis in which both dynamin1-mCherry and dynamin1-GFP are co-expressed (called DYNctrl in the text). We find that the control shows the same duration of events for both channels (Figure 4), shows 67% of coincident events (Figure 4), which is significantly different from the dyn1+dynab and dyn2+dynab (Figure 4). Indeed, as stated by the reviewer, we find that about 33% of the events in the control are not concomitant, which is most likely due to the noise in the signal and the analysis we are performing. Still, up to 65% of events are not concomitant in Dyn1+dynab (reversed statistics compared to the DYNctrl control), which statistically shows that dynab detects non-concomitant events.

3) It would be advisable that the authors test the uptake of a model cargo (transferrin) in the presence/absence of Dynab expression.

The point made by the reviewer is fair, and prompted us into performing transferrin uptake assays in cells overexpressing dynab. We obtained the very striking result that transferrin internalization is significantly lower when expressing dynab alone or in co-expression with dynamin 2 mCherry in HeLa cells (Figure 4—figure supplement 3). This was even more striking that we did not observe significant change of the CCP dynamics in cells transfected with dynab (see Figure 4). We thus concluded that even if the Clathrin-Coat machinery was not affected by dynab expression, the internalization of transferrin was impaired, most probably because its receptor was not properly internalized. Indeed, when overexpressing dynab, the Transferrin receptor (TfR) stays at the plasma membrane, and no endosomal pool is visible, as opposed to control cells (see Figure 4—figure supplement 3). We interpreted these results in the light of the reviewer’s hypothesis that dynab could create molecular crowding, blocking the specific interaction between TfR and the clathrin coat required for proper internalization of transferrin.

Even though puzzled by these results, we wondered if other cargoes suffered the same internalization defect. We thus checked the clathrin mediated internalization of EGF (using low concentrations of EGF, see: Sigismund S, Argenzio E, Tosoni D, Cavallaro E, Polo S, Di Fiore PP. Clathrin-mediated internalization is essential for sustained EGFR signaling but dispensable for degradation. Dev Cell. 2008 Aug;15(16):209-19). In this case, EGF uptake was not significantly affected (Figure 4—figure supplement 4), suggesting that the impact of dynab overexpression was specific to Transferrin. Even more striking, dynab overexpression had no effect when dyn1 was expressed, suggesting that the higher affinity of dynab for dyn2 could cause the impairment of TfR internalization.

The authors consider that these results, although striking, clearly show that when dynamin1 is used, dynab does not cause any noticeable CME defect, both at the cargo and at the coat level, strongly supporting our proposition that dynab detects the GTPase activity of dyn1 at the CPPs in conditions close to physiological conditions.

4) There are several issues with the analysis. First, and in a related vein to (16) it is striking that in general the kinetics of dynamin recruitment are consistently slower in the presence of Dynab (Figure 4) and perhaps the pair-wise comparisons used here will mask this global trend.

We point out that in Figure 4, the general trend seems to be that the kinetics of dynamin are faster (not slower) when dynab is overexpressed, even though non-significant using a P-test. The pair-wise comparisons used are in our opinion the only valid ones, since cell type and expressed dynamin isoform are the same, only dynab is co-expressed in one of the two conditions.

An ANOVA type analysis using all the data in Figure 4 would be more appropriate here.

The point of this figure was to compare the effect of dynab overexpression on duration of dynamin events in the different cell types transfected with dyn1 or dyn2. We do not think an ANOVA analysis would make the point clearer, as it tests the statistical relevance of differences observed in conditions where two or more parameters are changed (typically dynamin type and cell type).

Also the metric of 'time divided by time' has no units and the authors should use a different way to express this (perhaps using fractional notation).

We have calculated time difference (Δ*t*, time between the peak of dynab and the peak of dynamin) and have expressed them in percentage of the corresponding duration of dynamin peak, so the unit is in percentage of dynamin duration, which is similar to the fractional notation proposed by the reviewers.

[Editors' note: further revisions were requested prior to acceptance, as described below.]

After careful consideration and consultation between the reviewers we have reached a consensus. We have all agreed that the work reported in your manuscript is important and that your finding that dynab can detect the GTP loaded state of Dnm1 is new and important, and that this will prove a useful tool for other researchers. However, we have also agreed that your new finding that dynab blocks Tfn uptake (but not EGF uptake) is a key piece of information that the readers need to have at hand when they read the manuscript. Therefore, these data should be moved from their current position in the supplementary data to a figure in the main text. Also, we must ask for one further experiment – that is to measure coated pit lifetime in control cells versus cells expressing dynab. These results should be included with the Tfn/EGF data in one figure which evaluates the effects of dynab on endocytosis.

We have added the last experiment required by the reviewers, showing that the lifetime of clathrin-coated pits – visualized by imaging Clathrin light-chain-mCherry – did not change significantly upon dynab overexpression.

We also added Figure 5 and Figure 6 to show the interference of dynab with Transferrin uptake (Figure 5) and the unchanged uptake of EGF under dynab overexpression (Figure 6).